# "It's a slightly different vibe". New pathways in condition-specific rehabilitation for people with new or existing joint pain

Sheree A. McCormick[1], Jenny Alexanders[2], Gillian Yeowell [1], Francis Fatoye[1], Nigel Timothy Cable[3], Patrick Doherty[4], Davina Deniszczyc[2], Victoria Fitzgerald[2], Panayiotis Michael[1,2]*

1 Faculty of Health and Education, Manchester Metropolitan University, Manchester, United Kingdom, 2 Nuffield Health, Epsom, Surrey, United Kingdom, 3 Institute of Sport, Manchester Metropolitan University, Manchester, United Kingdom, 4 Department of Health Sciences, University of York, York, United Kingdom

* panayiotis.michael@stu.mmu.ac.uk

## Abstract

### Background

Musculoskeletal (MSK) conditions are a leading cause of pain and disability in adults. Exercise-based rehabilitation programmes are recommended however, sustained behaviour change is often poor. New rehabilitation pathways designed to promote adherence to exercise, can be iteratively developed using behaviour change models. This study explored the experiences of people living with joint pain participating in a Joint Pain Programme (JPP), a unique community-based rehabilitation initiative delivered by exercise professionals, that is offered 'free of charge' to patients and provides supplementary access to a local fitness and well-being centre. The findings are mapped to behaviour change models to inform implementation strategies that enhance exercise adherence in this population.

### Methods

A qualitative design, informed by pragmatism, using semi-structured interviews was used to explore participants' experiences of uptake and attendance at a 12-week community-based rehabilitation programme for joint pain. Findings were analysed using inductive thematic analysis. NVivo software was used to facilitate analysis, with models of behaviour change used to interpret the findings. The study is reported in accordance with the consolidated criteria for reporting qualitative (COREQ) research.

### Results

21 interviews took place online with individuals who attended the programme. Four themes were identified: 1) The programme supports my needs; 2) What motivates me; 3) The 'value add' environment; and 4) What hinders me from exercising.

**Data availability statement:** Data supporting this publication are available on the Open Science Framework: URL https://osf.io/kf5h7/files.

**Funding:** This research was funded through an industry-academic partnership between Nuffield Health, a not-for-profit organisation (England and Wales Charity Number: 205533; Scotland Charity Number: SC041793) and Manchester Metropolitan University (Grant/Award Number: Not Applicable). No specific grant was awarded to individual authors. Whilst senior representatives from both organisations contributed to the conception of the research and the preparation of the manuscript, the research team independently conducted data collection, data analysis, and interpretation of the findings, and retained full responsibility for the decision to publish.

**Competing interests:** SM (lead author), GY, FF, NTC and PD are not members of the Nuffield Health charity, the funding organisation, and as such declare no competing interests. All other contributors were employees of the Nuffield Health charity during preparation of this manuscript. The views expressed in this paper do not necessarily represent the views, decisions or policies of Nuffield Health charity. No financial relationships with any organisations that might have an interest in the submitted work in the previous three years have been declared. No other financial associations with authors' spouse or children under the age of 18 have been declared. DD is a trustee on the board of The National Academy of Social Prescribers. No other non-financial associations or activities that may be relevant to the manuscript have been declared. This does not alter our adherence to PLOS ONE policies on sharing data and materials.

## Conclusion

The JPP provides a new pathway for MSK rehabilitation that is perceived positively by people living with joint pain. Uptake and attendance in the early stage of exercise adoption is influenced by multiple interventions acting at the policy, community, organisational, interpersonal and intrapersonal level. Recommendations for policy and programme designers are made. The structure of the JPP could act as a potential springboard where programmes for other long-term conditions could be rolled out, reducing the burden on valuable health service resources.

## Introduction

Musculoskeletal (MSK) disorders such as widespread joint or back pain, gout, rheumatoid arthritis and osteoarthritis, are leading causes of pain and disability in adults, with the burden increasing in most countries [1–3]. The clinical presentation and progression of musculoskeletal disorders varies from person to person, with commonly reported symptoms including (but not limited to), joint pain, stiffness, muscular weakness, and impairments in mobility [4]. These ongoing symptoms lead to physically impaired quality of life (QoL) and poor emotional health [5,6]. Lifestyle modification including exercise and education are recognised as non-pharmacological cornerstones for pain and condition management [2,7]. Championing movement-based approaches to living well, the World Health Organisation calls for safe, accessible, affordable, and appropriate spaces to be physically active in their Global Action Plan on Physical Activity 2018–2030 [8], and stresses special attention be paid to vulnerable groups; i.e., people with disabilities and chronic conditions.

The number of movement-based rehabilitation programmes in the community is slowly growing [9]. For example, exercise referral schemes are commonly used in the National Health Service (NHS) for managing adults who are living with, or at risk of developing, a chronic disease [10]. Adults are typically referred into these 10–16-week exercise programmes by a GP or healthcare practitioner. Despite being in operation for over two decades, these services remain fragmented and have disjointed referral systems, creating significant pressure on NHS resources and barriers for patients [11]. Exercise professionals, employed in non-clinical settings, e.g., independent fitness settings, could help alleviate some of these issues. For example, specialist-trained exercise professionals providing rehabilitation with clinical oversight, could complement what is provided locally, thereby reducing the burden on overstretched NHS resources. In addition, fitness settings that provide leisure time physical activity alongside rehabilitation programmes could help people who have completed the rehabilitation programme experience a more cohesive transition from dependent, i.e., professional-led physical activity to independent fitness and lifetime physical activity [12].

Recently, the Nuffield Health Charity (Charity Numbers: England and Wales 205533; Scotland SC041793) established the delivery of a professionally led, condition-specific rehabilitation programme, underpinned by evidence and clinically

assured on a national scale [13]. Specialist trained exercise professionals, referred to as rehabilitation specialists (RS), delivered the Joint Pain Programme (JPP) at 110 community-based fitness and wellbeing centres. The JPP is 'free of charge' to patients and focusses on the management of osteoarthritis and other musculoskeletal (MSK) conditions, through physical activity, education and emotional wellbeing support. Facilitating ease of access through on-line self-referral, the JPP comprises 12-weeks of RS-led rehabilitation, consisting of 2 × 1-hour sessions per week (40 mins physical activity; 20 mins education), with a maximum of 12 participants in each group. The follow-on, independent phase comprises 12-weeks of supplementary access to a local fitness and wellbeing centre, also 'free of charge' to patients. Access to a bespoke app containing pre-recorded exercise sessions and health information webinars is also provided. After the first 24 weeks, attendees can opt to take up ongoing membership at the fitness and wellbeing centre at a subsidised rate. A service evaluation of the initial 12-week programme reported significant reduction in joint pain and stiffness, personal well-being and joint function [13]. Although this previous work has demonstrated patient benefit, the barriers, facilitators and motivational factors associated with programme uptake, attendance and lifelong fitness and physical activity (LFPA) are yet to be determined. Understanding these factors in the iterative development of the JPP may further improve and promote self-management skills and health outcomes for this population.

Frequently reported facilitators to uptake and attendance in exercise-referral schemes include social support (from providers, peers and family), behaviour regulation, and a variety of personalised exercise sessions [14]. Reported barriers typically include session time, location, financial cost, intimidating gym atmosphere and a lack of confidence in operating gym equipment [10]. Relatively little is known about the experiences of people with a chronic condition, e.g., an MSK condition, accessing community fitness centres for LFPA compared to other adults [15]. This is an important aspect to understand as exercise interventions for people with MSK, or other chronic conditions, do not typically promote sustained behaviour change [12,15]. In their recent review, Nikolajsen and colleagues [15] identified important factors influencing exercise uptake in fitness centre settings, including accessibility, financial costs, suitability of equipment and social support. The authors highlighted that adults with disability were more likely to experience negative feelings towards fitness centres as a barrier to exercise compared to adults without a disability. The review called for more research on the actual experience of people with disabilities at fitness centres, including psychological, social and environmental factors, to better understand the factors that influence uptake and adherence. Such research could inform development of services that seek to provide equitable rehabilitation, particularly as individuals living with MSK conditions often experience physical disabilities [15,16]. This could include shaping policy and guidelines for integrating programmes into NHS care pathways, informing funding decisions for preventative healthcare, or supporting the case for commissioning fitness-based interventions as part of long-term condition management.

Addressing these shortfalls, the current study captured the experiences of people living with long-term MSK conditions participating in the JPP. The aim of the project was to gain insight into the factors that influence programme uptake and attendance, providing recommendations for programme refinement and policymakers. Several behavioural theories underpinned this project. The Theoretical Domains Framework (TDF) was selected for its capacity to theoretically investigate the cognitive, affective, social and environmental influences on behaviour [17], which can be mapped onto specific intervention functions and behaviour change techniques [18]. This positions the TDF as an effective tool for identifying mechanisms through which the adoption and maintenance of exercise rehabilitation can be supported. However, some authors caution against the rigid application of the TDF framework, advocating for a more inductive and flexible approach to its utilisation, enabling the identification of factors beyond the pre-defined domains [19]. Given the community-embedded nature of the JPP, the Social Ecological Model (SEM) of health behaviours was also considered throughout the study [20]. Recent research employing the model has highlighted the importance of multi-level influences in promoting physical activity in people living with MSK pain, including patient-provider relationship, organisational resources, and access to opportunities for physical activity [21]. To further contextualise behavioural determinants over time, and highlight the dynamic nature of behaviour change, the Transtheoretical Model (TTM) was also incorporated within the study

[22]. The application of the model in previous physical activity and exercise intervention research [23–25], supports its relevance in guiding the interpretation of the findings for this project. We utilised the three behaviour change models, to provide a comprehensive yet adaptable theoretical foundation in our investigation and interpretation of findings which can effectively be applied to the pragmatic nature of the JPP.

## Methods

This study is reported in accordance with the consolidated criteria for reporting qualitative (COREQ) research [26]. Ethical approval was obtained from Manchester Metropolitan University Faculty Ethics Committee, UK (Ref: 59059). Informed consent was obtained from all participants verbally and digitally audio-recorded by the interviewer prior to participation.

### Study design

This study was nested in a longitudinal qualitative interview-based project exploring the sustainability of healthy behaviours following participation in a community joint pain programme. A qualitative design informed by pragmatism [27] was used to explore participants' experiences of participating in the JPP and generate meaningful and actionable results.

### Study population

A purposive sample of 21 participants were recruited between 8th January 2024 and 2nd February 2024 from cohorts that had recently participated in the JPP at five fitness and wellbeing centres in England, UK. A manager from the Nuffield Health charity acted as gatekeeper, informing RSs about the study and asking them to introduce the study to eligible individuals. The target sample size of 20–25 participants was based on recommendations for achieving data saturation when recruiting relatively homogenous participants from multiple sites [28]. Participants were eligible to participate in the study if they had participated in the 12-week JPP. To join the programme, participants were required to be aged 18 or over, have no planned surgery within 20 weeks or prior surgery in the last 12 weeks, and have experienced joint pain for at least six months. Participants were also required to have no uncontrolled medical conditions which was determined through a verbal dialogue between the participant and the RS following referral to the programme. A medical condition was deemed as controlled if participants reported they were aware of the condition, had previously received care from a healthcare professional for their condition, and were adhering to any prescribed medications or treatments. Additionally, participants were required to be able to visit one of the charity's fitness and wellbeing centres and access the online resources provided throughout the programme.

### Data collection

In-depth one to one semi-structured interviews were undertaken using Microsoft Teams or via telephone with a female member of the research team experienced in conducting qualitative interviews (SM). Participants were unknown to the interviewer, only the core research team (SM, PM, JA) had access to information that could identify individual participants. Participants were aware that they could withdraw at any point prior to analysis and all 21 participants completed the interviews. None of the participants required any repeat interviews. The interview schedule was informed by the Theoretical Domains Framework [17] (S1 File). New and unanticipated issues were probed when necessary, addressing previously raised concerns regarding rigidity in the utilisation of the TDF [19]. Interviews ranged between 45 and 70 minutes in duration and were digitally recorded and transcribed verbatim by the research team (SM and PM).

### Data analysis

Data analysis was undertaken using Braun and Clarke's six phase framework for thematic analysis [29]. SM and PM independently listened to the audio-recordings and read the transcripts line by line to identify salient text related to answering

the research question. Data was subsequently coded inductively and grouped to create sub-themes. Concordance on sub-themes was reached through discussion and critical review by SM and PM. Sub-themes were explored to discover over-arching patterns and main themes in the data. JA acted as a critical friend, sense-checking the over-arching patterns, main and sub themes [30]. There was agreement in the themes identified by the wider team and any refinement of themes related to semantics. Codes were recorded using NVivo software (version 14).

All participants received a thank you £10 gift voucher following their interview. All participants stated that they wished to be informed of the final report on the study. Member checking was used to inform the findings and ensure that participants' experience of the JPP were represented [26,31]. These participants did not receive incentives for their feedback.

## Results

Twenty-one participants were invited to participate, demographic information is provided in Table 1. All participants provided informed consent and attended the interview. Six participants reported holding membership at a Nuffield Health fitness and wellbeing centre in the past, and one participant had previously held a membership at an unnamed gym.

Saturation at ≤5% new information, was identified at 12$^{+2}$ interviews using a base of four and run of two, as per the method proposed by Guest and colleagues [32]. Additional data were collected beyond this point to account for potential drop-off in participation during the linked longitudinal study.

**Table 1. Demographic information.**

|     | Gender | Ethnicity | Age (Years) | Reason for referral | Time living with condition (Years) | Referral Pathway |
|-----|--------|-----------|-------------|---------------------|-----------------------------------|------------------|
| P1 | Female | White British | 65+ | Arthritis | 8 | Health Professional – Self-referral |
| P2 | Female | White Other | 65+ | Frozen shoulder | 4 | WOM – Self-referral |
| P3 | Male | White British | 65+ | Joint Pain | 10 | Social Media – Self-referral |
| P4 | Male | White British | 65+ | Arthritis | 4 | WOM – Self-referral |
| P5 | Male | White British | 65+ | Joint pain | 2 | WOM – Self-referral |
| P6 | Female | White British | 65+ | Osteoarthritis | 20 | WOM – Self-referral |
| P7 | Female | White British | 55-64 | Joint pain | 5 | Advert – Self-referral |
| P8 | Female | White British | 65+ | Both knees replaced | 10 | Social Media – Self-referral |
| P9 | Female | White British | 65+ | Both knees replaced | 4 | Social Media – Self-referral |
| P10 | Female | White British | 65+ | Joint pain | 20 | Social Media – Self-referral |
| P11 | Female | White British | 65+ | Joint pain | 10 | WOM – Self-referral |
| P12 | Female | White British | 65+ | Osteoarthritis | 10 | WOM – Self-referral |
| P13 | Female | White Irish | 65+ | Osteoarthritis and COPD | 9 | WOM – Health professional referral |
| P14 | Female | Indian | 65+ | Joint pain | 3 | Social Media – Self-referral |
| P15 | Female | British/Asian | 45-54 | Rheumatoid arthritis | 16 | Social Prescriber – Self-referral |
| P16 | Female | British/South Korean | 65+ | Knee and ankle pain (previous broken ankle) | 5 | WOM – Self-referral |
| P17 | Female | British | 65+ | Generalized arthritis | 3 | Health Professional – Self-referral |
| P18 | Female | White British | 65+ | Knee pain | 20 | Social Media – Self-referral |
| P19 | Female | White British | 55-64 | Arthritis & ME | 15 | WOM – Self-referral |
| P20 | Male | White British | 65+ | Osteoarthritis | 10 | Health Professional – Self-referral |
| P21 | Female | White British | 65+ | Knee pain | 12 | Health Professional – Self-referral |

COPD, Chronic obstructive pulmonary disease; ME, Myalgic encephalomyelitis; WOM, Word of Mouth. The mean age of participants was 69.0 years (SD = 7.84 years). The mean time living with a joint pain condition was 9.24 years (SD = 5.62 years).

Four overarching themes were identified within the data: 1) The programme supports my needs; 2) What motivates me; 3) The 'value add' environment; and 4) What hinders me from exercising. The hierarchy of themes and sub-themes are presented in Fig 1.

### Theme 1: The programme supports my needs

This theme captures how the JPP meets the multifaceted needs of individuals living with MSK conditions, who are seeking ways in which to improve their physical function and QoL.

**Graded group exercises build confidence and change beliefs.** Participants reported that the variety of exercises and graded approach, i.e., the flexibility to adapt exercises to their individual physical needs, facilitated participation. This approach alleviated participant concerns surrounding the safety of exercising and improved participants' physical activity self-efficacy.

*Umm, you know, being pushed, but not too much and the way it was done I was very impressed with. And they would slowly increase, you know, over the weeks.* [P11; Female; Joint Pain]

The opportunity to explore different types of exercise in a safe and supportive environment helped overcome initial intimidation, particularly in unfamiliar settings like the gym, where some participants felt uncertain about using fixed resistance machines.

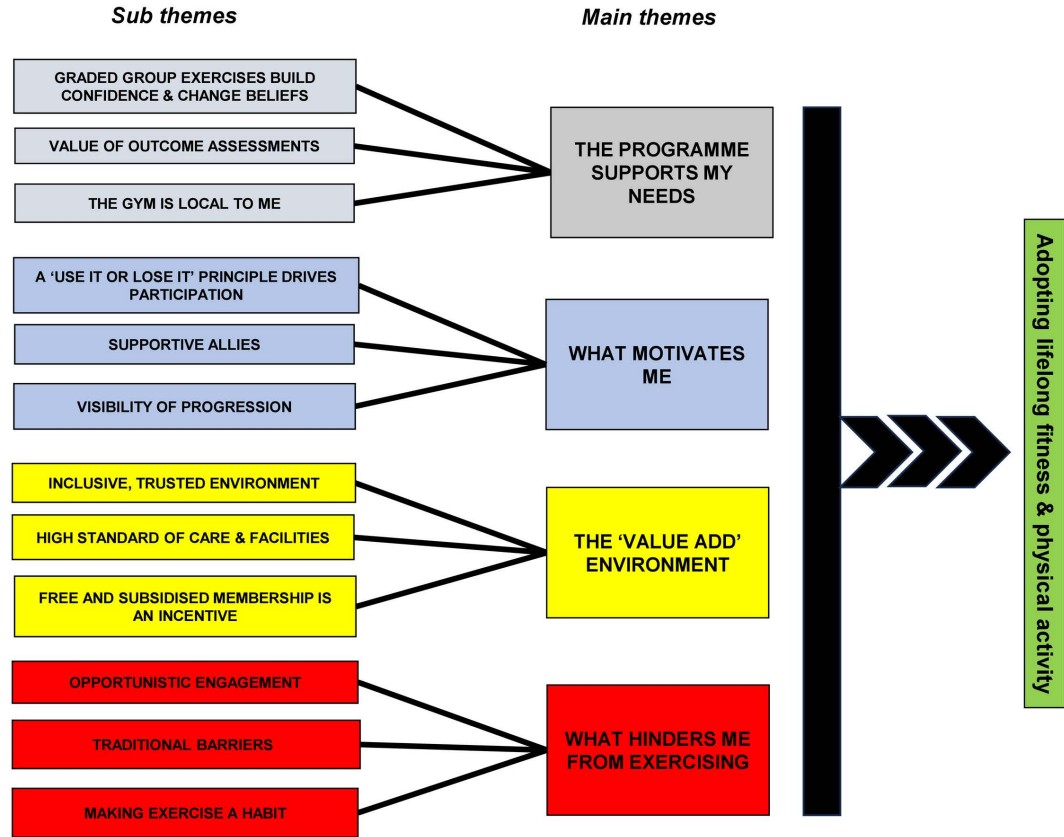

**Fig 1. Hierarchy of themes.**

*I don't feel comfortable going up to the gym, you know? I look at the sea of machines and think "Ohh, don't really know what I'm doing".* [P9; Female; Knee Replacement]

**Value of outcome assessments.** The majority of participants valued the outcome assessments, with some considering them as a superior service to that provided by GPs. For others, the assessments triggered negative emotions, yet helped regulate their behaviour. A minority of participants felt some of the outcome measures were not relevant to them as they were not actively looking to address those health factors.

*I was absolutely dreading it. So it really kept me on the straight and narrow in terms of exercising. Watching what I was eating at certainly cutting down on the booze.* [P8; Female; Knee Replacement]

Some participants were keen to understand more about the results of their outcome assessment, although these questions could not always be answered by the RS.

*It was a little bit weird because my blood sugar was higher at the end of the 12 weeks and it was at the beginning and neither [the RS] nor myself could figure out why.* [P16; Female; Knee and Ankle Pain]

**The gym is local to me.** The delivery of the JPP in a local fitness and wellbeing centre facilitated participation, with most participants reporting that a fifteen-minute drive time was acceptable. However, for participants reliant on public transport, accessibility and alignment of bus timetables and scheduled sessions required additional time and effort.

*I do seem to be going twice a week at least, and-, and very fortunate -it's literally a five minute drive away from me.* [P19; Female; Arthritis and Myalgic Encephalomyelitis]

### Theme 2: What motivates me

This theme encapsulates the primary motivations for participation and adherence to the supported phase of the programme.

**"A use it or lose it" principle drives participation.** Participants expressed differing motives for engaging in the JPP, with the majority indicating that they were primarily motivated by the negative impact of pain and loss of functional mobility on their daily lives. Other participants reported that their motivation was driven by the aim of avoiding the adverse consequences of physical inactivity, including further loss of mobility, exacerbation of pain, or need for surgical intervention.

*I couldn't get down on the ground easily and get up again. I found like I don't know, it was just before Christmas. I was doing a dress, sewing a hem. Oh, and I thought, how am I gonna get down to the bottom shelf?* [P11; Female; Joint Pain]

*Yeah, I mean, I've got to do down the route of knee replacement. Yet that's far too early. But the consultant would just say, well, let's do it. That's not the route I wanna go down.* [P21; Female; Knee Pain]

**Supportive allies.** Participants in the JPP appreciated the supportive network that developed between group members. The educational discussions fostered knowledge exchange and supported participants' emotional well-being, while helping them reframe their perceptions of pain. The RSs delivering the JPP were central to this network, facilitating discussions, listening, and providing encouragement and support.

*I think it made you feel like you're not alone with this problem. Because before this you think you know am I-, am I sort of-, am I the only one, you know, that's got this problem.* [P14; Female; Joint Pain]

Indirectly supporting the sub-theme of 'Supportive Allies', the majority of participants did not engage with the online resources (pre-recorded exercise sessions and health information webinars) accessed via a bespoke app. For the few that reported logging in, engagement was brief and infrequent. With respect to using the app for exercise, the general feeling was that there was a "*need to be doing it really around people in the right place rather than at home*" [P21; Female; Knee Pain].

**Visibility of progression.** Participant-led goal setting was supported by the RS. Some participants had experience of goal setting, some had never used goal setting before, and some preferred not to set goals because of fear of failure. Participants' goals were often pragmatic and closely linked to activities of daily life. This alignment provided a means of informal positive feedback on their progression during their everyday activities.

*…mine [my goals] were very simple to start off with purely because of the fact that you know I was walking on two crutches when I started so it was basically to walk without support and walk to the bus stop and then you know-, and the last one was to get on the bicycle. So I've kind of covered all those because I have been on the bicycle and I am walking to the bus stop.* [P16; Female; Knee and Ankle Pain]

Participants were also motivated through social exchange, whereby observing others perceived to be less physically able altered their perceptions of their own health severity. Additionally, witnessing others' improvements in physical activity further enhanced their motivation.

*I've been thrilled to see some of the people who at the beginning really struggled, who had really bad joint pain and who are now-, I see in the gym.* [P8; Female; Knee Replacement]

**Theme 3: The 'value add' environment**

This theme captures the positive, and sometimes unexpected, environmental factors that influenced participants' behaviour. Some participants were familiar with the charity but were surprised to learn of the 'free of charge' programmes.

**Inclusive, trusted environment.** The inclusive and welcoming environment of the fitness and wellbeing centre provided a space where participants felt a sense of belonging.

*So I think this is a-, it's a slightly different vibe...I think the thing that it gave me more than anything is the acceptance that there might be a place at the gym for me, you know, because think before, it didn't feel like my environment, I didn't feel like I belonged there.* [P19; Female; Arthritis and Myalgic Encephalomyelitis]

**High standard of care and facilities.** Many participants praised the high standard of care and facilities. The JPP filled a perceived void in primary care relating to the provision of, and access to, pain management services.

*No, I think the trainer actually gave the more-, had more knowledge than the GP. Yeah, I think it's because they spent more time with you. Whereas the GP didn't spend long enough. They don't have the time.* [P14; Female; Joint Pain]

**Free and subsidised membership is an incentive.** The 24-week 'free of charge to patients' policy incentivised participants to attend, particularly those on a pension.

*I mean, obviously the fact it's free a big incentive you know as well. Yet you know it's-, it's like, why wouldn't you do it if you're not able to do other things as much as you would like to, you know?* [P20; Male; Osteoarthritis]

**Theme 4: What hinders me from exercising**

This theme outlines the barriers to exercise adoption experienced by people living with joint pain, including factors beyond the control of the programme deliverers, as well as the skills participants need to regulate their own behaviour.

**Opportunistic engagement.** Despite the nationwide availability of the JPP, the majority of participants learned about the programme by chance through verbal conversations or serendipitously via social media. A small number were informed about the programme through healthcare professionals. There was no evidence of a clear or common pathway or signposting to the programme.

*[I heard about it from] one of my neighbours, UM, he's quite disabled and he heard about it through another friend. So he joined it, and then I heard about it through him.* [P11; Female; Joint Pain]

**Traditional barriers.** Inclement weather was a barrier for many, either preventing attendance or increasing reliance on cars or public transport. Other barriers included caregiving responsibilities, sessions being scheduled during working hours, and the impact of holiday travel.

*"…like next weekend, I've got my grandchildren coming so I won't go to the gym because that, you know, there's a ready-made excuse not to be able to go."* [P17; Female; Generalised Arthritis]

**Making exercise a habit.** Some participants described finding it difficult to make exercise a habit.

*Uhm, well, in some ways like doing it twice a week is really difficult, but in other ways like it's the best thing ever, because actually you do get into a routine.* [P15; Female; Rheumatoid Arthritis]

Some participants, however, were able to identify personal strategies they believed would support sustained exercise behaviour such as environmental cues and positive self-talk.

*I've done all the things I've advised students to do who struggled to begin to study. So, you know, getting the gym kit out the night before going when I get up. And the fact that it's on the corner, I can see that's really helping.* [P2; Female; Frozen Shoulder]

## Discussion

This study examined the experiences of individuals with MSK conditions participating in the JPP programme. Four themes emerged: 1) The programme supports my needs; 2) What motivates me; 3) The 'value add' environment; and 4) What hinders me from exercising. To go beyond descriptive analysis, we applied the SEM and TTM of behaviour change to the interpretation of our findings [20,22]. The SEM outlines how individual based environment interactions shape behaviour and serves as a framework to identify barriers to LFPA, including among those with disabilities [33]. The themes and sub-themes identified were mapped onto the SEM domains (Fig 2) to illustrate how these factors influenced JPP uptake and attendance. To examine changes across time at the cognitive and individual-level, we used the TTM, which highlights how multiple constructs interact to drive behaviour change [22,34]. These include six stages of change, ten processes of change (italicised throughout to indicate relevance), self-efficacy, and decisional balance [22,35].

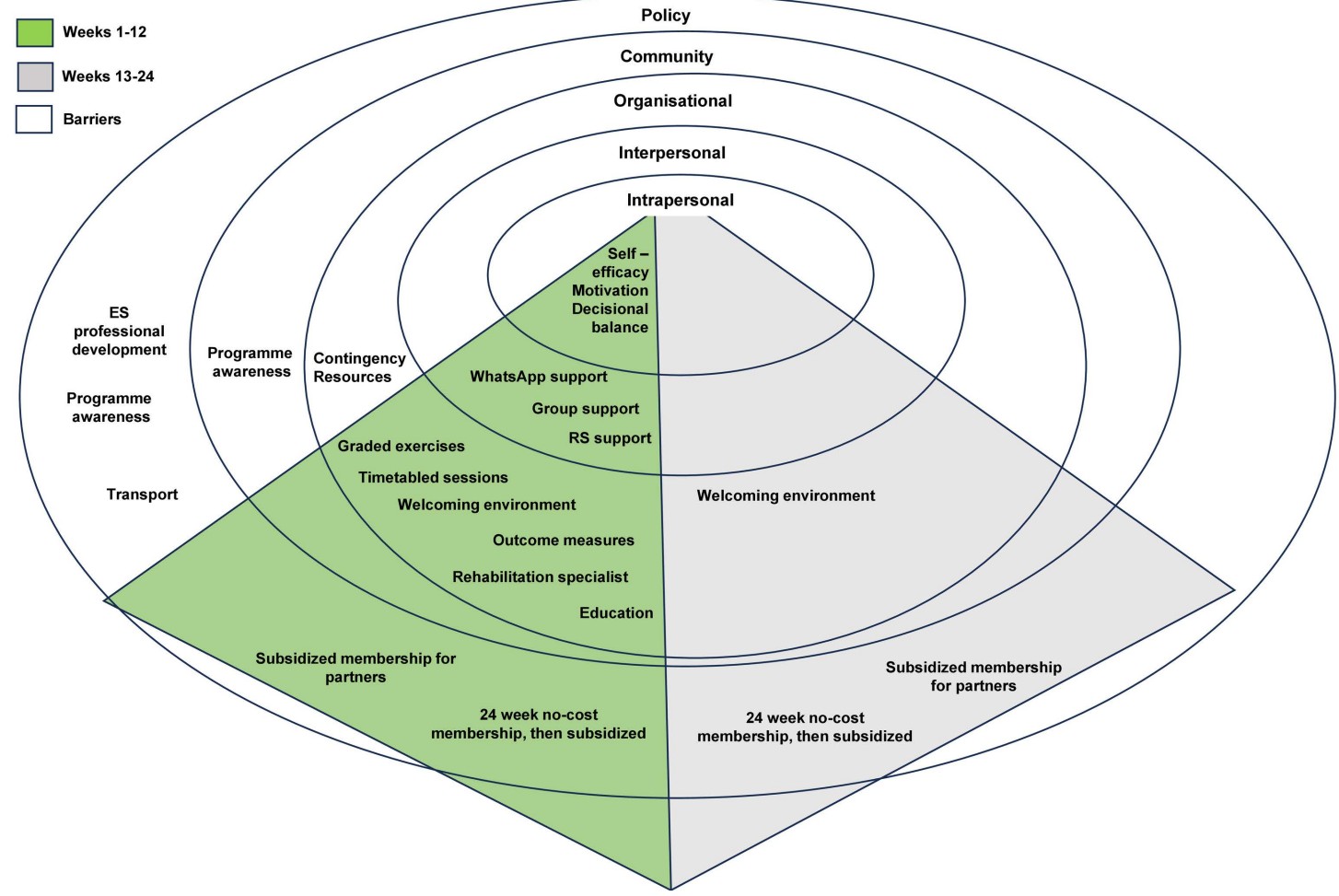

**Fig 2. Summary of results of participants' experiences situated within a Social Ecological Model.**

## The programme supports my needs

Participants described the JPP as timely and accessible, aligning with their readiness and motivation to address their functional limitations and pain. The online, self-referral JPP pathway acted as a straightforward 'call to action' (*self-liberation*), avoiding common referral barriers [11] and enabling progression from contemplation of exercise (contemplation stage of change construct) to preparation to exercise (preparation stage of change construct) [22,34].

The initial 12-week group-based education and exercise sessions employed *stimulus control* and *helping relationships* to support engagement [34]. The cardiorespiratory exercises were graded, ensuring sessions were inclusive and providing mastery experiences for all. These experiences were supported through RS positive feedback and encouragement *(reinforcement management)*. Observing others with lower functional ability, who were striving to achieve, inspired and motivated more functionally-able participants through vicarious experiences. Conversely, more able-bodied participants supported and encouraged those with poorer function, acting as role models. Mastery experiences, feedback, vicarious experiences and modelling are sources of self-efficacy [36], a key construct in the TTM [37], which leads to situation-specific confidence and supports behaviour change. However, psychological barriers emerged in resistance training sessions. Participants reported difficulty interpreting machine diagrams and expressed reduced self-efficacy, echoing prior studies

[10,33]. These barriers likely undermined decisional balance, despite resistance exercise being a primary intervention for improving MSK function [38]. Co-designed resistance training formats could mitigate such challenges, supporting exercise continuation into the action and maintenance phases of change, enhancing health outcomes and reducing system burden.

Outcome measures, a routine service evaluation process at the organisational level (Fig 2), were collected at the beginning and end of the 12-week programme. For many, results served as formal evidence of progress aiding self-efficacy and reinforcing behaviour change through self-re-evaluation and reinforcement management. These measures enabled RSs to monitor and provide tailored feedback [39]. Participants valued this formal feedback saying it *"kept them on track"* [P8; Female; Knee Replacement]. Indeed, some participants suggested that future programmes could provide formal, personalised feedback reports for participants to share with family and friends as evidence of their progress, fostering *helping relationships*. In terms of exercise adoption, these reports could also highlight successes and areas to work on to further improve participants' construct of self-efficacy [23,39]. These findings illustrate how standard data collection can actively shape behaviour across intrapersonal and interpersonal domains, reinforcing the need for intentional design of intervention components across levels of influence.

To effectively support participants, RSs should possess the interpersonal skills and knowledge to manage conversations around outcome measures or outcomes that are misunderstood. They should also be equipped to interpret physiological data (e.g., blood pressure) and its impact on motivation and emotional response. Training should emphasise the distinction between reinforcing behaviour progress and outcome achievement, with the former being more conducive to sustained behaviour change [40]. Behaviour change skills essential for RSs could be embedded into a formal accreditation pathway, encouraging wider adoption from both workforce and health professional perspectives [41].

Importantly the delivery model of the JPP in a non-clinical setting, increased choice and accessibility to exercise rehabilitation for individuals living with MSK conditions in the community. Consistent with previous findings [10,21,42,43], participants reported that the proximity of programme sites facilitated engagement and supported favourable decisional balances, with most willing to travel 15–30 minutes by car. Caution is warranted in interpreting our findings, as all interviewees had completed the programme and were, by necessity, able to attend twice-weekly sessions over 12 weeks. Notably, participants dependent on public transport reported that bus schedules often conflicted with JPP session times, potentially creating a less favourable decisional balance (Fig 2). These findings highlight the need for cross-sector collaboration, where session timings are coordinated with public transport schedules, and public transport availability is tailored to meet local health needs. Scaling up the delivery of the JPP through host organisations embedded in local communities may reduce transportation related barriers for certain individuals and contribute to enhancing local infrastructure and accessibility.

## What motivates me

At the second level of the SEM, the current study identified the importance of interpersonal factors, i.e., the support of group peers, the RS, and family members. Informal WhatsApp groups enabled participants to coordinate attendance with peers, fostering a shared commitment to change (Fig 2). Group education sessions further supported *helping relationships* as individuals exchanged experiences and coping strategies with others 'in the same boat'. In similar chronic pain contexts, such peer support has been shown to enhance self-efficacy, increase activity, and reduce pain and perceived disability [44]. Future research could explore how *helping relationships* exert their effect in the JPP and for whom – it may depend on how individuals view the group and may not always be positive [45]. This information could be used to develop strategies to optimise the effectiveness of this process of change activity, as individuals transition through the action and maintenance stage of change constructs to become lifelong independent exercisers [46–48].

Participants acknowledged the RSs' expertise in exercise and joint pain, which likely enhanced decisional balance. However, a minority perceived topics such as mindfulness and mental health as beyond the RSs' professional remit. As reported in other work [41,49], this finding suggests there is a need to understand and define the required skills and competencies of the RS role, to facilitate the professional development of this workforce.

The RSs conducted pragmatic, participant-led goal setting that tended to target activities of daily life. As such, this approach provided an incentive for participants, and in conjunction with behavioural practice, yielded mastery experiences away from the gym, as participants went about their daily lives [50]. This fostered self-efficacy, leading to a more desirable decisional balance. Goals that are specific, measurable, attainable, realistic and time-oriented (SMART) are reported to foster self-efficacy more than alternative goals [23]. Upskilling RSs to understand goal setting theory and effectively take a person through a goal setting process may further foster participants' construct of self-efficacy, leading to improved decisional balance and ultimately sustaining LFPA. However, further research is required to explore the effectiveness and utilisation of SMART goals within the context of symptom management for individuals with chronic conditions [51–53].

## The 'value add' environment

The delivery environment of the JPP emerged as a key determinant of participant experience and behaviour (Fig 2). At the intrapersonal level, the JPP provided important social resources (emotional support, information, access to new social contacts), essential aspects in social identity and self-efficacy, and important components of overall wellbeing [54]. At the organisational level, the servicescape (i.e., the visual, amenities, neatness and hygiene) of the fitness and wellbeing centre, created a favourable impression on participants, influencing perceptions of service quality [55]. At the policy level, offering zero-cost membership through a trusted community organisation incentivised participation and reinforced behaviour, addressing financial barriers commonly cited in exercise uptake [10,15,56]. Together, interventions across SEM levels helped cultivate not just a physical setting but a socially meaningful space promoting engagement and identity [55]. Importantly, the outcomes of the interventions at each level filtered across and exerted their effect at the intrapersonal level, influencing psychological factors such as relatedness, motivation, self-efficacy and decisional balance, ultimately supporting behaviour change. The positive impact of a no-cost membership at the intrapersonal level suggests that community-based rehabilitation centres should advocate for policies supporting sustained access for low resource populations. Policymakers might also evaluate the cost effectiveness of lifestyle interventions in reducing long-term healthcare expenditures. The JPP's impact on physical, mental and social well-being has enabled a contribution of over £86.5 million in social value to the UK economy in 2023, by facilitating participants to return to employment and improve their health outcomes [57].

## What hinders me from exercising

Our findings identified limited programme awareness at the community level as a barrier to uptake. Participants typically became aware of the programme opportunistically mainly via word of mouth, less so through social media, and rarely from clinicians. Similar awareness related barriers to community rehabilitation engagement have been noted elsewhere [11]. Awareness could be improved by designing interventions to strengthen relationships across community groups or partnering with community disability organisations [33]. External policymakers can also play a vital role to help raise awareness of community rehabilitation programmes and embed these into NHS care pathways. Greater integration of such programmes into the NHS pathways could improve access and uptake particularly for underserved populations [57], thereby improving health outcomes, reducing health inequalities and reducing healthcare resource utilisation.

In agreement with others [10,58,59], the results of this study highlight several practical barriers to participation in the fitness and wellbeing programme, with weather conditions, caregiving responsibilities and work commitments emerging as significant obstacles for some. These findings suggest that while environmental factors, such as weather, are often overlooked in discussions about health interventions, they can significantly influence participation [58]. Although beyond the direct control of programme designers, these barriers should be addressed by incorporating alternative delivery modes that enable participant contingency planning [33]. Providing targeted support for participants with caregiving

responsibilities, advocating for improved transportation options, and offering alternative modes of delivery (e.g., live online sessions), are options to explore [60]. Addressing these challenges across various levels of the SEM (e.g., policy and organisation) is essential for improving access to exercise rehabilitation for individuals living with MSK conditions, particularly in areas with more variable weather or limited transport infrastructure.

All participants intended to continue exercising post-programme, with some using personal strategies (*stimulus control*), while others struggled to establish routine. Evidence supports a reciprocal link between self-efficacy and habit formation each reinforcing the other [61,62]. This raises concerns about whether participants, once in the LFPA phase, will maintain exercise habits without continued social and organisational support (see Fig 2, weeks 13–24). This is an area we will explore in the longitudinal study.

### Limitations and strengths

This study highlights several practice-based strengths. For example, we have identified the key factors that contribute to the success of a programme for this population, providing a crucial first step in the development of new pathways for joint pain rehabilitation. The pragmatic design of the study further supports the translation of these findings to similar settings, with the implementation recommendations more likely to lead to positive outcomes in real-world applications. A limitation of this study is that physical activity/ JPP attendance data was not collected and only discussed through interviews. Collecting this data through objective tools may have had some influence on the interpretation of our findings. A second limitation is that the majority of participants were female, older adults (retired or semi-retired) who had self-referred. Thus, the recommendations made are relevant to this predominantly homogenous group and may not apply to under-served populations.

### Conclusion

The community-based, 12-week JPP meets the short-term rehabilitation needs of people living with joint pain in a unique manner – delivered locally by an exercise professional, with supplementary access to a fitness and wellbeing centre that can support LFPA. By employing the TDF, SEM and TTM, our findings demonstrate how human behaviour is influenced by the dynamic interaction between the individual, the environment, and psychological constructs such as self-efficacy, social support, and processes of change. Drawing on these identified factors, we have provided recommendations for programme designers and policymakers to improve access to interventions for people with MSK conditions. Additionally, these considerations could serve as a foundation for adopting and testing similar models of rehabilitation in other chronic conditions and potentially alleviate the burden on the NHS.

### Key recommendations

The JPP has shown that RSs can safely and effectively deliver these new models of rehabilitation to the satisfaction of people with long-term conditions. Given the identified importance of self-efficacy across various processes of change and habit formation, the JPP should aim to address identified barriers and actively foster the development of self-efficacy throughout the intervention. This includes providing participants with opportunities for mastery experiences with gym equipment, reinforcing participant efforts via RS feedback, and optimising goal setting to support sustained behaviour change throughout activities of daily life.

There is also a need to raise awareness of, and actively support, movement-based initiatives that are 'free of charge' to patients. These efforts should be promoted at the policy, organisational and community level to provide people living with joint pain greater choice in management of their condition. This would not only improve the physical and mental health outcomes across communities, but it would also contribute to measurable social value and economic benefits for the UK. Embedding community-based movement rehabilitation into NHS pathways is recommended to help support the NHS in meeting its growing demands.

## Supporting information

**S1 File. Interview schedule.**
(DOCX)

**S2 File. COREQ checklist.**
(DOCX)

**S3 File. Email sent to participants for member checking.**
(DOCX)

## Acknowledgments

We extend our gratitude to the participants in this study, whose contributions were essential to the completion of this work. We also thank Pooja Kumari for reviewing and supporting the translation of findings into policy recommendations.

For the purpose of open access, the author(s) has applied a Creative Commons Attribution (CC BY) licence to any Author Accepted Manuscript version arising from this submission.

## Author contributions

**Conceptualization:** Sheree A. McCormick, Gillian Yeowell, Francis Fatoye, Nigel Timothy Cable, Patrick Doherty, Davina Deniszczyc.

**Data curation:** Sheree A. McCormick, Panayiotis Michael.

**Formal analysis:** Sheree A. McCormick, Jenny Alexanders, Panayiotis Michael.

**Funding acquisition:** Nigel Timothy Cable, Davina Deniszczyc.

**Investigation:** Sheree A. McCormick.

**Methodology:** Sheree A. McCormick.

**Project administration:** Sheree A. McCormick, Panayiotis Michael.

**Resources:** Sheree A. McCormick, Panayiotis Michael.

**Software:** Sheree A. McCormick, Panayiotis Michael.

**Supervision:** Gillian Yeowell, Francis Fatoye, Nigel Timothy Cable.

**Validation:** Jenny Alexanders.

**Visualization:** Sheree A. McCormick, Jenny Alexanders, Panayiotis Michael.

**Writing – original draft:** Sheree A. McCormick, Jenny Alexanders, Panayiotis Michael.

**Writing – review & editing:** Sheree A. McCormick, Jenny Alexanders, Gillian Yeowell, Francis Fatoye, Nigel Timothy Cable, Patrick Doherty, Victoria Fitzgerald, Panayiotis Michael.

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
