## [Decision Letter · Decision Letter 0]

6 Jun 2025

Dear Dr. Michael,

We look forward to receiving your revised manuscript.

Kind regards,

Md. Feroz Kabir, BPT, MPT, MPH, BPED, MPED

Academic Editor

PLOS ONE

**Journal Requirements:**

1. When submitting your revision, we need you to address these additional requirements. Please ensure that your manuscript meets PLOS ONE's style requirements, including those for file naming. The PLOS ONE style templates can be found at https://journals.plos.org/plosone/s/file?id=wjVg/PLOSOne_formatting_sample_main_body.pdf and https://journals.plos.org/plosone/s/file?id=ba62/PLOSOne_formatting_sample_title_authors_affiliations.pdf 2. In the online submission form, you indicated that your data will be submitted to a repository upon acceptance.  We strongly recommend all authors deposit their data before acceptance, as the process can be lengthy and hold up publication timelines. Please note that, though access restrictions are acceptable now, your entire minimal dataset will need to be made freely accessible if your manuscript is accepted for publication. This policy applies to all data except where public deposition would breach compliance with the protocol approved by your research ethics board. If you are unable to adhere to our open data policy, please kindly revise your statement to explain your reasoning and we will seek the editor's input on an exemption. 3. When completing the data availability statement of the submission form, you indicated that you will make your data available on acceptance. We strongly recommend all authors decide on a data sharing plan before acceptance, as the process can be lengthy and hold up publication timelines. Please note that, though access restrictions are acceptable now, your entire data will need to be made freely accessible if your manuscript is accepted for publication. This policy applies to all data except where public deposition would breach compliance with the protocol approved by your research ethics board. If you are unable to adhere to our open data policy, please kindly revise your statement to explain your reasoning and we will seek the editor's input on an exemption. Please be assured that, once you have provided your new statement, the assessment of your exemption will not hold up the peer review process. 4. We note that this data set consists of interview transcripts. Can you please confirm that all participants gave consent for interview transcript to be published? If they DID provide consent for these transcripts to be published, please also confirm that the transcripts do not contain any potentially identifying information (or let us know if the participants consented to having their personal details published and made publicly available). We consider the following details to be identifying information:- Names, nicknames, and initials- Age more specific than round numbers- GPS coordinates, physical addresses, IP addresses, email addresses- Information in small sample sizes (e.g. 40 students from X class in X year at X university)- Specific dates (e.g. visit dates, interview dates)- ID numbers Or, if the participants DID NOT provide consent for these transcripts to be published:- Provide a de-identified version of the data or excerpts of interview responses- Provide information regarding how these transcripts can be accessed by researchers who meet the criteria for access to confidential data, including:a) the grounds for restrictionb) the name of the ethics committee, Institutional Review Board, or third-party organization that is imposing sharing restrictions on the datac) a non-author, institutional point of contact that is able to field data access queries, in the interest of maintaining long-term data accessibility.d) Any relevant data set names, URLs, DOIs, etc. that an independent researcher would need in order to request your minimal data set. For further information on sharing data that contains sensitive participant information, please see: https://journals.plos.org/plosone/s/data-availability#loc-human-research-participant-data-and-other-sensitive-data If there are ethical, legal, or third-party restrictions upon your dataset, you must provide all of the following details (https://journals.plos.org/plosone/s/data-availability#loc-acceptable-data-access-restrictions):a) A complete description of the datasetb) The nature of the restrictions upon the data (ethical, legal, or owned by a third party) and the reasoning behind themc) The full name of the body imposing the restrictions upon your dataset (ethics committee, institution, data access committee, etc)d) If the data are owned by a third party, confirmation of whether the authors received any special privileges in accessing the data that other researchers would not havee) Direct, non-author contact information (preferably email) for the body imposing the restrictions upon the data, to which data access requests can be sent?

**Additional Editor Comments:**

Please submit the revised manuscript according to the reviewers' comments.

Reviewers' comments:

Reviewer's Responses to Questions

**Comments to the Author**

1. Is the manuscript technically sound, and do the data support the conclusions?

Reviewer #1: Yes

Reviewer #2: Yes

2. Has the statistical analysis been performed appropriately and rigorously?

Reviewer #1: N/A

Reviewer #2: No

3. Have the authors made all data underlying the findings in their manuscript fully available?

Reviewer #1: Yes

Reviewer #2: Yes

4. Is the manuscript presented in an intelligible fashion and written in standard English?

Reviewer #1: Yes

Reviewer #2: Yes

**Reviewer #1: ** The manuscript is well written and sets out a clear rationale for this study. Below are a few comments:

1. Authors should consider being specific about the study condition. Abstract background describes MSKs but introduction describes osteoarthritis. MSK conditions encompass a wide range of musculoskeletal conditions and thus will be a better background description for the study. Further, only four participants reported osteoarthritis as reason for referral.

2. Line 65 - Suggest using UK National Health Service in the first instance for global readers.

3. Well written introduction but authors do not provide any rationale for using the three behavioural theories. Why was the TDF used to inform interviews? Why was the TTM and SEM used to interpret findings? How does using these behavioural theories add to the gap in literature? What other behavioural theories have been used to explore this research area?

4. Line 124-15 Unclear what cross-sectional study is referred to here

5. Line 135-136 A clear definition for 'have no uncontrolled medical conditions' is required. One participant is reported having COPD. How do the authors justify that this is not an uncontrolled medical condition? How is an uncontrolled medical condition assessed?

6. Line 185 Suggest 'A use it or lose it principle' rather than approach

7. Line 185-196 Authors may consider adding this sub theme to Theme 2 given that participants describe being motivated by a concern of losing physical function.

8. Results - Suggest additional descriptors for quotes to aid reader. E.g., [P(number), gender, MSK condition] rather than P(number)

9. Line 214 Suggest 'Value of outcome assessments' as title instead.

10. Line 252 Period missing at end of sentence

11. Discussion is too long winded. An approach may be to discuss the key findings (rather than each theme/subtheme) within the context of the two frameworks. The SEM diagram is detailed enough to help reduce some of the text in this section. Authors should consider making reference to the diagram. The discussion should only highlight the key findings within the context of the literature.

12. Key recommendations should include recommendations for the JPP intervention. The findings have explored some barriers to engagement with the intervention. Recommendations to therefore improve attendance can be highlighted here.

**Reviewer #2: ** The title should be like this: "Condition-specific rehabilitation for people with new or existing joint pain. Need to describe the details of study procedures. Results should be the results, not only written on the theme. The inside of the theme will be presented as results or findings. The strength, weakness, and limitation should be addressed properly. Table 1: Age and Year with conditions should descriptively address where mean and SD should be addressed. English grammar correction is important.

**Do you want your identity to be public for this peer review?** For information about this choice, including consent withdrawal, please see our Privacy Policy

Reviewer #1: No

Reviewer #2: **Yes: ** Mohammad Mohinul Islam

---

## [Author Response · Author response to Decision Letter 1]

21 Jul 2025

We would like to thank the editor and reviewers for their thoughtful and constructive feedback, which has been instrumental in improving the quality of this manuscript. We have carefully considered each of the comments and have made the corresponding revisions, which we believe has strengthened the clarity, depth, and overall impact of the paper. Please note that any signposting to the relevant sections of the manuscript is in reference to the tracked changed version. We have also made some minor formatting amendments such as displaying the ages as age ranges to better protect the anonymity of participants. The below is also available as an attached document to this submission.

Editors’ comments

1. Please ensure that your manuscript meets PLOS ONE's style requirements.

Authors’ Response: Thank you for this comment. We believe the manuscript now meets these requirements.

2. In the online submission form, you indicated that your data will be submitted to a repository upon acceptance. We strongly recommend all authors deposit their data before acceptance, as the process can be lengthy and hold up publication timelines. Please note that, though access restrictions are acceptable now, your entire minimal dataset will need to be made freely accessible if your manuscript is accepted for publication. This policy applies to all data except where public deposition would breach compliance with the protocol approved by your research ethics board. If you are unable to adhere to our open data policy, please kindly revise your statement to explain your reasoning and we will seek the editor's input on an exemption.

Authors’ Response: Thank you for emphasising the importance of open data. We intend to upload the minimal dataset to a repository and plan to make the DOI publicly available upon acceptance of the manuscript. Please note, one participant did not consent to including their data in an open repository, therefore their data will not be included.

Authors’ Response: Thank you for this comment. We believe this comment relates to comment 2 and as such, our response is similar. We intend to upload the minimal dataset to a repository and plan to make the DOI publicly available upon acceptance of the manuscript.

4. We note that this data set consists of interview transcripts. Can you please confirm that all participants gave consent for interview transcript to be published? If they DID provide consent for these transcripts to be published, please also confirm that the transcripts do not contain any potentially identifying information (or let us know if the participants consented to having their personal details published and made publicly available).

Authors’ Response: All but one participant provided consent for the anonymised dataset to be published. The data from this participant will not be included. We have reviewed the intended dataset for publication to ensure no identifiable information will be included.

Reviewer 1 Comments

Reviewer 1: The manuscript is well written and sets out a clear rationale for this study. Below are a few comments:

1. Authors should consider being specific about the study condition. Abstract background describes MSKs, but introduction describes osteoarthritis. MSK conditions encompass a wide range of Musculoskeletal conditions and thus will be a better background description for the study. Further, only four participants reported osteoarthritis as reason for referral.

Authors’ Response: Thank you for your comment. The introduction section of the manuscript has included more global reference to MSKs disorders which also included OA as being one of those named conditions.

2. Line 65 – Suggest using UK National Health Service in the first instance for global readers

Authors’ Response: Thank you for your comment. This has been changed.

3. Well written introduction, but authors do not provide any rationale for using the three behavioural theories. Why was the TDF used to inform interviews? Why was the TTM and SEM used to interpret findings? How does these behavioural theories add to the gap in the literature? What other behavioural theories have been used to explore this research area?

Authors’ Response: Thank you for your comment. In response, we have clarified the rationale for the selection and application of each behavioural model within the Introduction.

4. Line 124- Unclear what cross-sectional study is referred to here

Authors’ Response: Thank you for your comment. We have changed the description of this to give it more clarity.

5. Line 135-136 A clear definition for ‘have no uncontrolled medical conditions’ is required. One participant is reported having COPD. How do the authors justify that this is not an uncontrolled medical condition? How is an uncontrolled medical condition assessed?

Authors’ Response: Thank you for raising this important aspect. We have now included some additional details around how a controlled / uncontrolled medical condition is assessed on pages 8 and 9 (line 168-173).

6. Line 185 suggest ‘A use it or lose it principle’ rather than approach

Authors’ Response: Thank you for your comment. We agree that this term better captures the essence of the sub-theme. The change can be seen on page 12.

7. Line 185-196 Authors may consider adding this sub theme to Theme 2 given that participants describe being motivated by a concern of losing physical function

Authors’ Response: Thank you for this comment, we believe this improves the flow of the paper and consistency in the thematic definitions. We have actioned this suggestion accordingly. In line with this change, the quote related to motivation through social exchange has also been moved to Theme 2 (Page 16 line 376).

8. Results – suggest additional description for quotes to aid reader e.g. (P (number), gender MSK condition etc rather than P(number)

Authors’ Response: Thank you for this comment. We have now included additional descriptors alongside quotes.

9. Line 214 – Suggest ‘value of outcome assessments’ as title instead

Authors’ Response: Thank you for your comment. We agree the wording improves the clarity of the sub-theme. The change can be seen on page 13 and reflected in the Discussion also.

10. Line 252 – Period missing at the end of sentence

Authors’ Response: Thank you for highlighting this. A period has been added (page 15 line 305).

11. Discussion is too long winded. An approach may be to discuss the key findings (rather than each theme/ subtheme) within the context of the two frameworks. The SEM diagram is detailed enough to help reduce some of the text in this section. Authors should consider making reference to this diagram. The discussion should only highlight the key findings within the context to the literature.

Authors’ Response: Thank you for your comment. The discussion section has been substantially revised to enhance clarity and precision. Key points have been restructured and rearticulated to provide greater focus and coherence throughout.

12. Key recommendations should include recommending for the JPP intervention. The findings have explored some barriers to engagement with the intervention. Recommendations to therefore improve the attendance can be highlighted here

Authors’ Response: Thank you for this comment. We have now included recommendations specific to improving outcomes and attendance of the JPP intervention.

Reviewer 2 Comments

1. The title should be like this: ‘Condition-specific rehabilitation for people with new or existing joint pain’

Authors’ Response: Thank you for your feedback. We believe the structure, duration, and level of support provided by the JPP represent a novel pathway that warrants clear distinction. While we acknowledge the presence of similar programmes in some UK communities, these typically do not incorporate the same structured format, educational components, or digital signposting offered by the JPP. Moreover, the specific expertise and training of Rehabilitation Specialists (RSs) is integral to the programme's delivery and are unique to this organisational context.

2. Results should be the results, not only written on the theme. The inside of the theme will be presented as results or findings.

Authors’ Response: Thank you for your observation. In line with conventional qualitative reporting, we structured the results section thematically, illustrating each theme with participant quotes to substantiate and contextualise our interpretations. These themes were then critically explored in the discussion section, where we developed and contextualised them in relation to existing literature and theoretical frameworks.

3. The strength, weakness and limitation should be addressed properly.

Authors’ Response: Thank you for your comments. We have aimed to present the strengths and limitations of the study in an honest and balanced manner to ensure transparency and support the rigour of our reporting. We appreciate your feedback and have reviewed this section again to confirm that it accurately reflects the scope and context of the work.

4. Table 1: Age and year with conditions should descriptively address where mean and SD should be addressed.

Authors’ Response: Thank you for your comment. The mean and SD of participant age and year living with conditions has been added to the Note. of Table 1.

5. English grammar correction is important.

Authors’ Response: Thank you for your comment. The manuscript has been thoroughly proofread to improve grammatical accuracy and punctuation, ensuring clarity and consistency throughout the sections.

---

## [Editor Report · Decision Letter 1]

25 Jul 2025

Dear Dr. Michael, 

Thank you for submitting your manuscript to PLOS ONE. After careful consideration, we feel that it has merit but does not fully meet PLOS ONE’s publication criteria as it currently stands. Therefore, we invite you to submit a revised version of the manuscript that addresses the points raised during the review process.

Please submit your revised manuscript by Sep 08 2025 11:59PM. If you will need more time than this to complete your revisions, please reply to this message or contact the journal office at plosone@plos.org . A rebuttal letter that responds to each point raised by the academic editor and reviewer(s). You should upload this letter as a separate file labeled 'Response to Reviewers'.A marked-up copy of your manuscript that highlights changes made to the original version. You should upload this as a separate file labeled 'Revised Manuscript with Track Changes'.An unmarked version of your revised paper without tracked changes. You should upload this as a separate file labeled 'Manuscript'.

We look forward to receiving your revised manuscript.

Kind regards,

Md. Feroz Kabir, PhD, BPT, MPT, MPH, BPED, MPED

Academic Editor

PLOS ONE

Journal Requirements:

**Additional Editor Comments:**

Yes, now the manuscript can be accepted after the English correction and format correction are properly done.

---

## [Author Response · Author response to Decision Letter 2]

1 Aug 2025

We would like to thank the editor for reviewing the revised manuscript and providing further feedback. We have reviewed comments and have made corresponding revisions where required. Please note that any signposting to the relevant sections of the manuscript is in reference to the tracked changed version. Our response is also available as an attachment to the online submission.

Journal Requirements:

Authors’ Response: Thank you for this comment. We believe we have only included relevant citations throughout the document. The reviewers did not provide any recommendations to cite specific work in their response.

Authors’ Response: Thank you for emphasising the importance of justifying the use of retracted papers. We have reviewed the references we have included using both EndNote and manually on Retraction Watch and can confirm that none are retracted.

Additional Editor Comments:

1. Yes, now the manuscript can be accepted after the English correction and format correction are properly done.

Authors’ Response: Thank you for this comment. We have identified formatting errors such as incorrect use of quotation marks (p15 line 298; p25 line 534-535) and additional space between words (p25 line 519-520). We have also identified two formatting issues within the reference list which we have now addressed (p26 line 551 and 568). We have removed a duplicate reference which appeared as a new citation (p5 line 91 and p27 line 586-588) and have updated the corresponding numbers to all other references and in text citations which follow. We have also ensured all journals within the references align where possible with the NLM abbreviations. Lastly, we have made one change to the author affiliations.

Authors’ Response: Thank you for reminding us of the tool. We have uploaded our figures to generate the correct formatting requirements, with the new files being provided with the re-submission.

---

## [Editor Report · Decision Letter 2]

24 Aug 2025

Thank you for submitting your manuscript to PLOS ONE. After careful consideration, we feel that it has merit but does not fully meet PLOS ONE’s publication criteria as it currently stands. Therefore, we invite you to submit a revised version of the manuscript that addresses the points raised during the review process.

Please submit your revised manuscript by next Oct 08 2025 11:59PM If you will need more time than this to complete your revisions, please reply to this message or contact the journal office at plosone@plos.org . A rebuttal letter that responds to each point raised by the academic editor and reviewer(s). You should upload this letter as a separate file labeled 'Response to Reviewers'.A marked-up copy of your manuscript that highlights changes made to the original version. You should upload this as a separate file labeled 'Revised Manuscript with Track Changes'.An unmarked version of your revised paper without tracked changes. You should upload this as a separate file labeled 'Manuscript'.

We look forward to receiving your revised manuscript.

Kind regards,

Md. Feroz Kabir, PhD, BPT, MPT, MPH, BPED, MPED

Academic Editor

PLOS ONE

Journal Requirements:

Additional Editor Comments:

Please submit the manuscript after thorough English correction within the next 15 days.

---

## [Author Response · Author response to Decision Letter 3]

29 Aug 2025

We would like to thank the editor for reviewing the revised manuscript and providing further feedback. We have reviewed comments and have made corresponding revisions where required. Please note that any signposting to the relevant sections of the manuscript is in reference to the tracked changed version.

Journal Requirements:

Authors’ Response: Thank you for this comment. We believe we have only included relevant citations throughout the document. The reviewers did not provide any recommendations to cite specific work in their response.

Authors’ Response: Thank you for emphasising the importance of justifying the use of retracted papers. We have reviewed the references we have included using both EndNote and manually on Retraction Watch and can confirm that none are retracted. We have also ensured our reference list is complete and correct.

Additional Editor Comments:

1. Please submit the manuscript after thorough English correction within the next 15 days.

Authors’ Response: Thank you for this comment. We have carefully reviewed the manuscript for grammar, punctuation and clarity, and have made necessary English language corrections throughout the document:

Abstract (lines 25-27; 35-36)

Introduction (lines 51-52; 54; 56-57; 79-80; 92; 96; 100; 104)

Methods (lines 157; 162; 181-182; 191)

Results (lines 201-202; 241; 270; 276; 283-284; 302; 304-305; 329)

Discussion (lines 360; 375; 379; 392; 398; 421-422; 427; 438; 443; 446; 468; 503-504; 517)

Conclusion (lines 520-521)

Key Recommendations (lines 542-544)

---

## [Editor Report · Decision Letter 3]

26 Oct 2025

“It’s a slightly different vibe”. New pathways in condition-specific rehabilitation for people with new or existing joint pain

PONE-D-25-13162R3

Dear Dr. Michael,

We’re pleased to inform you that your manuscript has been judged scientifically suitable for publication and will be formally accepted for publication once it meets all outstanding technical requirements.

Kind regards,

Md. Feroz Kabir, PhD, BPT, MPT, MPH, BPED, MPED

Academic Editor

PLOS ONE
---

## [Editor Report · Acceptance letter]

PONE-D-25-13162R3

PLOS ONE

Dear Dr. Michael,

I'm pleased to inform you that your manuscript has been deemed suitable for publication in PLOS ONE. Congratulations! Your manuscript is now being handed over to our production team.

Kind regards,

on behalf of

Dr. Md. Feroz Kabir

Academic Editor

PLOS ONE